# Active pulmonary tuberculosis and coronavirus disease 2019: A systematic review and meta-analysis

**Ashutosh Nath Aggarwal** *, **Ritesh Agarwal, Sahajal Dhooria, Kuruswamy Thurai Prasad, Inderpaul Singh Sehgal, Valliappan Muthu**

Department of Pulmonary Medicine, Postgraduate Institute of Medical Education and Research, Chandigarh, India

* aggarwal.ashutosh@outlook.com

## Abstract

### Objective

The proportion of COVID-19 patients having active pulmonary tuberculosis, and its impact on COVID-19 related patient outcomes, is not clear. We conducted this systematic review to evaluate the proportion of patients with active pulmonary tuberculosis among COVID-19 patients, and to assess if comorbid pulmonary tuberculosis worsens clinical outcomes in these patients.

### Methods

We queried the PubMed and Embase databases for studies providing data on (a) proportion of COVID-19 patients with active pulmonary tuberculosis or (b) severe disease, hospitalization, or mortality among COVID-19 patients with and without active pulmonary tuberculosis. We calculated the proportion of tuberculosis patients, and the relative risk (RR) for each reported outcome of interest. We used random-effects models to summarize our data.

### Results

We retrieved 3,375 citations, and included 43 studies, in our review. The pooled estimate for proportion of active pulmonary tuberculosis was 1.07% (95% CI 0.81%-1.36%). COVID-19 patients with tuberculosis had a higher risk of mortality (summary RR 1.93, 95% CI 1.56–2.39, from 17 studies) and for severe COVID-19 disease (summary RR 1.46, 95% CI 1.05–2.02, from 20 studies), but not for hospitalization (summary RR 1.86, 95% CI 0.91–3.81, from four studies), as compared to COVID-19 patients without tuberculosis.

### Conclusion

Active pulmonary tuberculosis is relatively common among COVID-19 patients and increases the risk of severe COVID-19 and COVID-19-related mortality.

**Data Availability Statement:** All relevant data are within the paper and its Supporting Information files.

**Funding:** The authors received no specific funding for this work.

**Competing interests:** The authors have declared that no competing interests exist.

## Introduction

The ongoing coronavirus disease 2019 (COVID-19) pandemic is spreading relentlessly, and has affected more than 230 million people worldwide. COVID-19 is associated with worse outcomes in the elderly population, and those with comorbid health conditions such as obesity, diabetes mellitus, hypertension, and cardiovascular disorders [1–9].

Tuberculosis is a destructive pulmonary disease and therefore widely perceived to be associated with increased susceptibility to acquiring COVID-19, and poorer prognosis in patients having both diseases concurrently, especially among people living with human immunodeficiency virus infection (PLHIV). A note from World Health organization (WHO) also anticipated poorer outcomes in these patients [10]. However, the actual impact of tuberculosis on occurrence and clinical outcomes of COVID-19 is not clear. A case series from Italy reported a benign clinical course for patients having both infections [11]. An early meta-analysis of six Chinese studies found no association between tuberculosis and COVID-19 severity or mortality [12]. Population based data from South Korea also did not suggest tuberculosis to be significantly associated with COVID-19-related mortality [13]. However, other investigators describe a disproportionately higher rate of adverse clinical outcomes among patients with tuberculosis and COVID-19 [14–16]. Tuberculosis was identified as the commonest comorbidity on verbal autopsy among 70 COVID-19 deaths, and in 10% of whole-body autopsies, in Zambia [17, 18]. Two meta-analyses suggest higher odds of underlying tuberculosis among patients with severe COVID-19 and those dying from COVID-19 [19, 20]. Due to these inconsistencies, we felt a need to perform a detailed analysis of the available evidence till date. Herein, we evaluate the frequency of concurrent active pulmonary tuberculosis among COVID-19 patients. We also assess if comorbid pulmonary tuberculosis increases the risk of severe disease, hospitalization, or mortality in COVID-19 patients.

## Methods

We registered our systematic review protocol with the PROSPERO database (registration number CRD42021245835). We followed the Preferred Reporting Items for Systematic Reviews and Meta-Analyses (PRISMA) and the Meta-analysis of Observational Studies in Epidemiology (MOOSE) recommendations for reporting our review [21, 22]. An approval from our Institutional Review Board was not necessary as we extracted only summary information from previously published articles.

### Search strategy

We initially looked up the PubMed and EMBASE databases for publications indexed till March 31, 2021, and further updated our search on June 30, 2021. We queried the PubMed database using the following search string: (Tuberculosis OR Tubercular OR Tuberculous OR TB OR Mycobacterium OR Mycobacterial) AND (COVID-19 OR "COVID 19" OR COVID19 OR nCoV OR 2019nCoV OR 2019-nCoV OR CoV-2 OR "CoV 2" OR SARS-CoV-2 OR SARSCoV2). The Embase database was similarly searched. We further scanned the WHO compendium of tuberculosis/COVID-19 studies for any additional published studies [23]. We also examined the bibliographies of selected articles and recent reviews.

### Selection of studies

After removing duplicate citations, two reviewers (ANA and RA) screened all the titles and abstracts. We omitted publications not reporting on COVID-19 or tuberculosis. We also excluded experimental, radiological or autopsy studies, case reports, letters to editor not

describing original observations, reviews, guidelines, conference abstracts, editorials, and study protocols. Full texts of citations considered potentially suitable by either reviewer were assessed further.

We included a publication for data synthesis if it (a) included patients with COVID-19 confirmed by detection of novel severe acute respiratory syndrome coronavirus 2 (SARS-CoV-2) RNA in respiratory specimens, or strongly suspected on clinical or radiological assessment if a confirmatory test was not available, (b) either described the frequency of patients having concurrent active pulmonary tuberculosis among COVID-19 patients, or reported on any of the following outcomes in COVID-19 patients with and without tuberculosis—severe COVID-19, hospital admission, or mortality. Severe COVID-19 was defined based on institutional or national guidelines, or as per the prevalent guidance from international professional bodies or the World Health Organization. If the same (or substantially overlapping) patient cohort was reported in two or more publications, we included the one describing the largest patient population. In case of any disagreement, consensus between the two reviewers determined study inclusion.

## Data extraction and study quality

We obtained information on study design, location and healthcare setting, participant inclusion and exclusion criteria, period of patient enrollment, the source of patient information, and the outcomes reported, from all eligible studies. We used the Newcastle-Ottawa Scale (NOS) to assess methodological quality of studies [24]. We considered a study to be of good quality if its NOS score was seven or more (out of a maximum possible score of nine).

## Statistical analysis

We estimated the percentage of active tuberculosis patients among those with COVID-19 disease in each study and calculated the corresponding 95% confidence interval (95% CI) by Clopper-Pearson exact method [25]. We also computed the relative risk (RR), and the corresponding 95% CI, for each predefined outcome from each study [26]. We employed a continuity correction of 0.5 for studies having 'zero' cell frequencies prior to these calculations.

We pooled our data using the DerSimonian-Laird random effects model to generate summary estimates [27]. Freeman-Tukey double arcsine transformation was used to summarize data on proportions [28]. We assessed between-study heterogeneity through the Higgins' inconsistency index ($I^2$), which was considered high for values greater than 0.75 [29]. The contribution of each study to overall heterogeneity, and its influence on the summary estimate, were assessed through the Baujat's plot [30]. For searching the reasons for heterogeneity, we performed subgroup analyses for predefined covariates that included study location and setting, study design, COVID-19 diagnostic standards, description of criteria used to define active tuberculosis, national burden of tuberculosis, and the overall study quality. World Health Organization standards were used to refer to countries as high burden, and to extract country incidence estimates, for tuberculosis [31]. In a sensitivity analysis, the influence of each study on the summary estimate was also assessed by repeating meta-analysis after iteratively omitting one study at a time. Further, any influential study was identified using a battery of diagnostic tests using Studentized Residuals, Difference in Fits (DFFITS), Cooks Distance, Covariance Ratio, Tau Square, and the contribution of each study in the Q, H2 test statistics value and the weights assigned to these studies [32]. Publication bias was assessed through Eggers' test and by visualizing contour-enhanced trim-and-fill funnel plots [33, 34]. We utilized the statistical softwares Stata (Intercooled edition 12.0, Stata Corp, USA) and R (version 4.1.1, R Foundation for Statistical Computing, Austria) for analyzing our data.

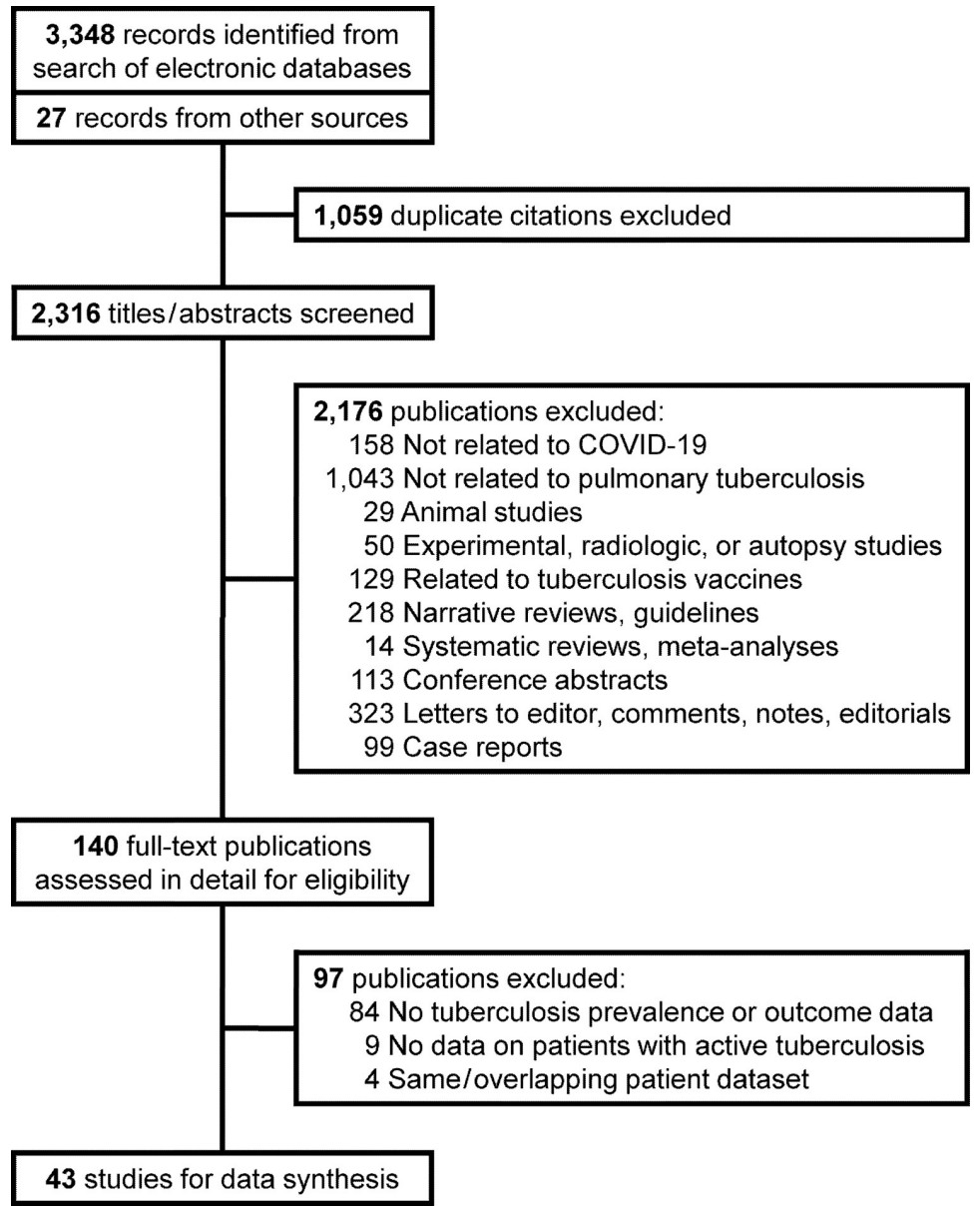

**Fig 1. Flow chart for study selection.**

## Results

We identified 3,375 publications from our literature search (Fig 1). We finally selected 43 studies, describing 236,863 patients with COVID-19, for data synthesis [35–77]. Thirty-three (76.7%) of them provided information on one or more of the adverse clinical outcomes of interest (Table 1). There were 30 (69.8%) publications from Asia, and 10 (23.3%) from Africa, with maximum contribution from China (22 studies) (Table 1). One (2.3%) study analyzed data from multiple countries [43]. All studies evaluated data from retrospective patient cohorts, except for four (9.3%) that collected the information prospectively [39, 40, 56, 69]. Six (14.0%) studies reported population-based data [35, 36, 42, 52, 59, 62], while the others were conducted in a hospital setting. Two (4.7%) studies also included COVID-19 patients based on

**Table 1.  Characteristics of included studies.**

| Author, year | Location | Study design | Setting | Inclusion criteria for COVID-19 patients | Exclusion criteria | Study period | Source of data | Tuberculosis definition | No. of COVID-19 patients | PLHIV | Information reported | NOS score |
|---|---|---|---|---|---|---|---|---|---|---|---|---|
| Al Kuwari HM, 2020 [35] | Qatar | Retrospective | General population | Confirmed disease | NS | Feb 28—Apr 18, 2020 | National COVID database | ICD codes | 5,685 | NS | Proportion, severity | 6 |
| Boulle A, 2020 [36] | Western Cape, South Africa | Retrospective | General population | Confirmed disease in adult patients (> = 20 years) | NS | Mar 1—Jun 9, 2020 | Health records database | Microbiological confirmation, anti-tubercular treatment, admission to tuberculosis hospital | 22,308 | 17.8% | Proportion, hospitalization, mortality | 8 |
| Chen T, 2020 [37] | Wuhan, China | Retrospective | Inpatients | Confirmed disease | NS | Jan 1—Feb 10, 2020 | Medical records | Medical records | 203 | 1.0% | Proportion, mortality | 6 |
| Dai M, 2020 [38] | China | Retrospective | Inpatients | Confirmed disease in patients with at least two CT examinations and discharged by study end date | Poor quality CT scan images | Feb 5—Mar 8, 2020 | Medical records | NS | 73 | NS | Proportion, severity | 5 |
| Du RH, 2020 [39] | Wuhan, China | Prospective | Inpatients | Confirmed or highly probable disease | NS | Dec 25, 2019—Feb 7, 2020 | Medical records | NS | 179 | NS | Proportion, mortality | 6 |
| Gupta N, 2020 [40] | New Delhi, India | Prospective | Inpatients | Confirmed disease | NS | Mar 20—May 8, 2020 | Medical records | NS | 200 | NS | Proportion | 5 |
| Ibrahim OR, 2020 [41] | Katsina, Nigeria | Retrospective | Inpatients | Confirmed disease in adult patients (> = 18 years) | NS | Apr 10—Jun 10, 2020 | Medical records | Confirmation during hospital stay | 45 | NS | Proportion, mortality | 6 |
| Lee SG, 2020 [42] | South Korea | Retrospective | General population | Confirmed disease in adult patients (> = 18 years) | NS | Mar 26—May 15, 2020 | Health insurance database | Diagnostic codes | 7,339 | 0.1% | Proportion, severity, mortality | 7 |
| Li G, 2020 [43] | 59 countries in China, North America, and Europe | Retrospective | Inpatients | Confirmed disease | Patients receiving remdesivir or dexamethasone, lack of treatment records, data from countries with <5 records | Jan 1—Apr 30, 2020 | Medical records | NS | 598 | NS | Proportion, mortality | 7 |

(*Continued*)

**Table 1.** (Continued)

| Author, year | Location | Study design | Setting | Inclusion criteria for COVID-19 patients | Exclusion criteria | Study period | Source of data | Tuberculosis definition | No. of COVID-19 patients | PLHIV | Information reported | NOS score |
|---|---|---|---|---|---|---|---|---|---|---|---|---|
| Li X, 2020 [44] | Wuhan, China | Ambispective | Inpatients | Confirmed or highly probable disease | NS | Jan 26 –Feb 5, 2020 | Personal/ telephonic interviews, medical records | ICD-10 diagnostic codes | 548 | NS | Proportion, severity | 6 |
| Liu J, 2020 [45] | Wuhan, China | Retrospective | Outpatients and inpatients | Confirmed disease in adult patients (> = 18 years) | NS | Dec 29, 2019—Feb 28, 2020 | Medical records | Medical records | 1,190 | NS | Proportion, hospitalization, mortality | 6 |
| Liu SJ, 2020 [46] | Ezhou, China | Retrospective | Inpatients | NS | NS | Jan 23 –Feb 12, 2020 | Medical records | NS | 342 | NS | Proportion, severity | 5 |
| Ma Y, 2020 [47] | 9 Chinese provinces | Retrospective | Inpatients | Confirmed disease in adult patients (> = 18 years) | NS | Jan 13 –Apr 13, 2020 | Medical records | Self-reported or diagnosed at admission | 1,160 | NS | Proportion, severity, mortality | 6 |
| Maciel EL, 2020 [48] | Espirito Santo, Brazil | Retrospective | Inpatients | Confirmed disease in patients with definite outcomes (discharge or death) | NS | Feb 26 –May 14, 2020 | Regional epidemiologic studies database | NS | 440 | 1.0% | Proportion, mortality | 5 |
| Nachega JB, 2020 [49] | Kinshasa, DR Congo | Retrospective | Inpatients | Confirmed disease | Incomplete information | Mar 10 –Jul 31, 2020 | Medical records | NS | 766 | 1.6% | Proportion, severity | 6 |
| Parker A, 2020 [50] | Cape Town, South Africa | Retrospective | Inpatients | Confirmed disease in adult patients (> = 18 years) | NS | Mar 24–11 May, 2020 | Medical records | NS | 113 | 21.2% | Proportion, hospitalization | 5 |
| Sun Y, 2020 [51] | Beijing, China | Retrospective | Inpatients | Confirmed disease | NS | NS | Medical records | NS | 63 | NS | Proportion | 5 |
| Sy KTL, 2020 [52] | Philippines | Retrospective | General population | Confirmed disease | NS | Up to May 17, 2020 | National COVID-19 surveillance registry | History or current diagnosis of tuberculosis | 12,513 | NS | Proportion, hospitalization, mortality | 8 |
| Xiao KH, 2020 [53] | Chongqing, China | Retrospective | Inpatients | Confirmed disease | NS | Jan 23 –Feb 8, 2021 | Medical records | NS | 143 | NS | Proportion, severity | 5 |
| Yu HH, 2020 [54] | Wuhan, China | Retrospective | Inpatients | Confirmed disease | NS | Jan 27 –Mar 5, 2020 | Medical records | NS | 1561 | NS | Proportion, severity | 7 |

*(Continued)*

**Table 1.** (Continued)

| Author, year | Location | Study design | Setting | Inclusion criteria for COVID-19 patients | Exclusion criteria | Study period | Source of data | Tuberculosis definition | No. of COVID-19 patients | PLHIV | Information reported | NOS score |
|---|---|---|---|---|---|---|---|---|---|---|---|---|
| Zeng JH, 2020 [55] | Shenzhen, China | Retrospective | Inpatients | Confirmed disease | NS | Jan 11—Apr 1, 2020 | Medical records | NS | 416 | NS | Proportion, severity | 5 |
| Zhang JJ, 2020 [56] | Wuhan, China | Prospective | Inpatients | Confirmed disease | NS | Jan 16—Feb 3, 2020 | Medical records | NS | 140 | NS | Proportion, severity | 6 |
| Zhang YT, 2020 [57] | Guangdong, China | Retrospective | Inpatients | Confirmed disease | NS | Jan 15—Mar 4, 2020 | Disease surveillance database | NS | 1,350 | NS | Proportion, severity | 5 |
| Abraha HE, 2021 [58] | Mekelle, Ethiopia | Retrospective | Inpatients | Confirmed disease | NS | May 10—Oct 16, 2020 | Medical records | NS | 2,617 | 0.9% | Proportion, severity | 7 |
| Dave JA, 2021 [59] | Western Cape, South Africa | Retrospective | General population | Confirmed disease | NS | Mar 4—Jul 15, 2020 | Regional health information database | Database records | 64,476 | 12.3% | Proportion | 6 |
| du Bruyn, 2021 [60] | Cape Town, South Africa w | Retrospective | Inpatients | Confirmed disease | NS | Jun 11—Aug 28, 2020 | Medical records | Microbiologically proven or clinically diagnosed | 104 | 29.8% | Proportion, severity | 6 |
| Gaibhiye RK, 2021 [61] | Mumbai, India | Retrospective | Inpatients | Confirmed disease in pregnant/postpartum women | NS | Apr—Sep, 2020 | Medical records | NS | 879 | NS | Proportion | 4 |
| Hesse R, 2021 [62] | South Africa | Retrospective | General population | Confirmed disease in adult patients (> = 18 years) | Indeterminate COVID-19 test results | Mar 1—Jul 7, 2020 | Health records | GeneXpert positivity within six months before COVID diagnosis | 98,335 | 6.3% | Proportion | 6 |
| Kapoor D, 2021 [63] | New Delhi, India | Retrospective | Inpatients | Confirmed disease in children <18 years age | NS | Mar 1—Dec 31, 2020 | Medical records | NS | 120 | NS | Proportion, mortality | 4 |
| Lagrutta L, 2021 [64] | Buenos Aires, Argentina | Retrospective | Outpatients and inpatients | Confirmed disease | NS | Jul 5—Oct 17, 2020 | Hospital registry | Bacteriological confirmation or recent clinical diagnosis | 5,447 | 7.2% | Proportion | 6 |
| Li S, 2021 [65] | Wuhan, China | Retrospective | Inpatients | Confirmed disease in patients with definite outcomes (discharge or death) | Patients still hospitalized at study end date, death within 24 hours after admission, loss to follow up | Jan 18—Mar 29, 2020 | Medical records | NS | 2,924 | NS | Proportion, mortality | 4 |

(Continued)

Table 1. (Continued)

| Author, year | Location | Study design | Setting | Inclusion criteria for COVID-19 patients | Exclusion criteria | Study period | Source of data | Tuberculosis definition | No. of COVID-19 patients | PLHIV | Information reported | NOS score |
|---|---|---|---|---|---|---|---|---|---|---|---|---|
| Lu Y, 2021 [66] | Wuhan, China | Retrospective | Inpatients | Confirmed severe disease in adult patients (<65 years) | NS | Jan 25—Feb 15, 2020 | Medical records | NS | 77 | NS | Proportion, mortality | 5 |
| Meng M, 2021 [67] | Wuhan, China | Retrospective | Inpatients | Confirmed severe disease | NS | Jan 2—Mar 28, 2020 | Medical records | NS | 415 | None | Proportion, mortality | 5 |
| Mithal A, 2021 [68] | New Delhi, India | Retrospective | Inpatients | Confirmed disease in adult patients (> = 18 years) | NS | Jul 9—Aug 8, 2020 | Medical records | NS | 401 | NS | Proportion | 5 |
| Moolla MS, 2021 [69] | Cape Town, South Africa | Prospective | Inpatients | Confirmed disease | NS | Mar 26—Aug 31, 2020 | Medical records | NS | 363 | 14.6% | Proportion | 6 |
| Song J, 2021 [70] | Wuhan, China | Retrospective | Inpatients | Confirmed disease in patients with definite outcomes (discharge or death) | NS | Feb 1—Mar 6, 2020 | Medical records | Self-report on admission | 961 | NS | Proportion, severity | 6 |
| van der Zalm MM, 2021 [71] | Cape Town, South Africa | Retrospective | Inpatients | Confirmed disease in children (< = 13 years) | Infants born diagnosed in the neonatal service, multisystem inflammatory syndrome | Apr 17—Jul 24, 2020 | Medical records | Medical records | 159 | 1.3% | Proportion, mortality | 5 |
| Verma R, 2021 [22] | Firozabad, India | Retrospective | Inpatients | Confirmed disease among critically ill patients in ICU | Patients referred to other centres | Jul 1—Dec 31, 2020 | Medical records | NS | 120 | NS | Proportion | 4 |
| Yan B, 2021 [73] | Jilin, China | Retrospective | Inpatients | Confirmed disease | NS | Jan 28—Mar 25, 2020 | Medical records | NS | 190 | NS | Proportion, severity | 5 |
| Yang C, 2021 [74] | Taiyuan, China | Retrospective | Inpatients | Confirmed disease | NS | Jan 24—Apr 25, 2020 | Medical records | NS | 104 | NS | Proportion, severity | 6 |
| Yitao Z, 2021 [75] | Guangzhou, China | Retrospective | Inpatients | Confirmed disease | NS | Jan 21—Mar 23, 2020 | Medical records | NS | 257 | NS | Proportion, severity | 5 |

(Continued)

**Table 1.** (Continued)

| Author, year | Location | Study design | Setting | Inclusion criteria for COVID-19 patients | Exclusion criteria | Study period | Source of data | Tuberculosis definition | No. of COVID-19 patients | PLHIV | Information reported | NOS score |
|---|---|---|---|---|---|---|---|---|---|---|---|---|
| Zhang W, 2021 [76] | Taiyuan, China | Retrospective | Inpatients | Confirmed disease | Patients with malignant tumors, hypertension, heart disease, diabetes, etc. | Jan 1—May 31, 2020 | Medical records | NS | 500 | NS | Proportion | 5 |
| Zheng B, 2021 [77] | Honghu, China | Retrospective | Inpatients | Confirmed disease | Laboratory and radiology workup at other hospitals, no pulmonary lesion of chest CT scan | Jan 1—Mar 27, 2020 | Medical records | NS | 198 | NS | Proportion, severity | 5 |

COVID-19 Coronavirus disease 2019, ICD International Classification of Diseases, NOS Newcastle-Ottawa Scale for study quality, NS Not specified, PLHIV People living with human immunodeficiency virus infection.

high clinical or radiological suspicion [39, 44]. All others only studied patients with disease confirmed by the detection of SARS-CoV2 RNA in respiratory specimens. One (2.3%) study did not specify the inclusion criteria [46]. Only two (4.7%) studies specifically evaluated children [63, 71]; others included only adults or described a mixed population. Patient information was retrieved mainly from medical records at participating healthcare facilities, or from surveillance registries or insurance databases (Table 1). Most investigators reviewed patient records or used tuberculosis-related diagnostic codes in databases to identify patients having active tuberculosis (Table 1). Fourteen (32.6%) studies reported human immunodeficiency virus (HIV) seroprevalence in their patient cohorts [36, 37, 42, 48–50, 58–60, 62, 64, 67, 69, 71]. Of these, a single study from South Africa provided tuberculosis prevalence and outcome data based on HIV status [36]. Only six (14.0%) studies were considered high quality (S1 Table) [36, 42, 43, 52, 54, 58].

## Proportion of patients with active tuberculosis

The proportion of patients having active pulmonary tuberculosis among COVID-19 patients could be computed from all 43 studies. It ranged from 0.18% to 14.42% (Fig 2). The highest occurrence was noted in a study conducted in a high HIV prevalence South African setting [60]. All other studies described figures below 6%. Almost all studies reported proportion estimates of comorbid pulmonary tuberculosis among COVID-19 patients that were higher than their corresponding WHO country estimates for annual tuberculosis incidence (Fig 2). The pooled proportion estimate from all 43 studies was 1.07% (95% CI 0.81%-1.36%).

There was substantial heterogeneity between the studies ($I^2$ 94.7%). Baujat's plot suggested that three studies unduly influenced heterogeneity as well as pooled estimates (S1 Fig) [36, 59, 62]. Omitting these three studies from meta-analysis resulted in a slightly higher summary estimate of proportion (1.15%, 95% CI 0.84%-1.50%) with only a minor reduction in heterogeneity ($I^2$ 86.9%). On influence analysis, a single study with the highest reported proportion of active tuberculosis patients was associated with large values of Studentized residuals, Cook's distance and DFFITS (S2 Fig), and was considered potentially influential [60]. After removing this study, the pooled proportion estimate from remaining 42 studies was lower at 1.00% (95% CI 0.75%-1.28%) with hardly any reduction in heterogeneity ($I^2$ 94.5%). On sensitivity analysis, omitting other studies one at a time also did not appreciably influence summary estimates or heterogeneity (S3 Fig). On subgroup analysis, studies conducted in low tuberculosis burden or multiple countries showed lesser heterogeneity (Table 2). Overall, the pooled estimates on proportion were much lower from studies conducted in countries not having high tuberculosis burden, as well as from population-based and high-quality studies (Table 2).

## Severe COVID-19

Twenty studies with 24,371 COVID-19 patients, of whom 161 (0.7%) had tuberculosis, provided information on severe COVID-19 [35, 38, 42, 44, 46, 47, 49, 51, 53–58, 70, 73–77]. All, except four (20.0%), of these publications were from China [35, 42, 49, 58]. Severe COVID-19 was defined based on World Health Organization guidance in three (15.0%) studies [35, 49, 58], recommendations from international professional bodies in two (10.0%) studies [44, 70], national guidelines in 11 (55.0%) studies [38, 42, 46, 47, 51, 53, 56, 57, 73, 74, 76], and institutional policy in four (20.0%) studies [54, 55, 75, 77]. Only three (15.0%) studies were considered high quality [42, 54, 58]. All studies, except two, included patients with laboratory confirmed COVID-19 [44, 46]. Only one (12.5%) had a prospective study design [56]. Of the 3431 patients with severe disease in the included cohorts, 36 (1.0%) had underlying tuberculosis. Only four (20.0%) studies reported a RR for severe COVID-19 that significantly exceeded

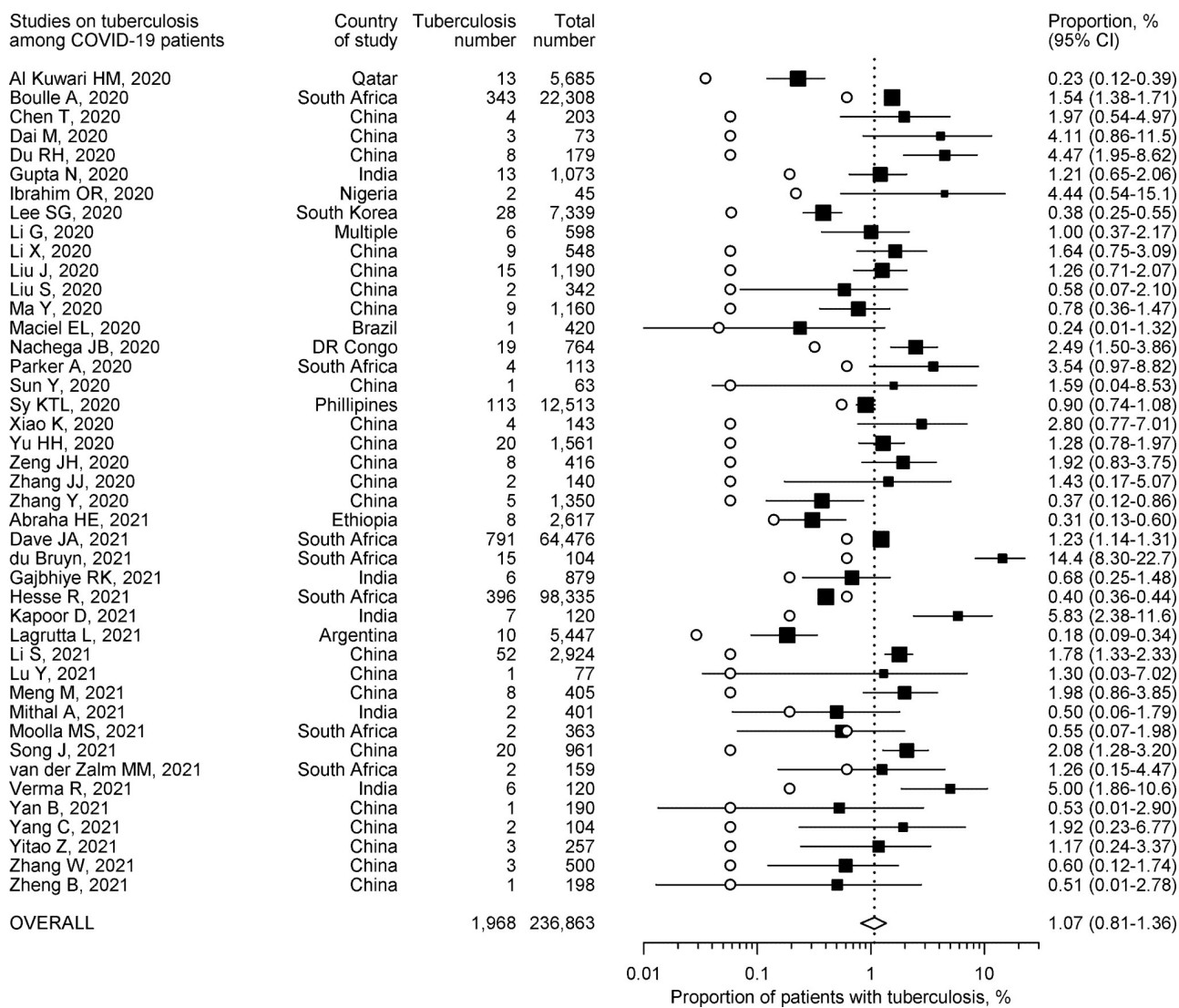

**Fig 2. Proportion of COVID-19 patients also having tuberculosis and corresponding 95% confidence intervals (CI).** Individual proportion estimates are depicted by solid squares, and the corresponding country estimate of annual tuberculosis incidence by hollow circles.

1.0 (Fig 3) [42, 46, 56, 77]. COVID-19 patients who also had tuberculosis were 1.46 (95% CI 1.05–2.02) times more likely to develop severe COVID-19 as compared to COVID-19 patients without tuberculosis (Fig 3).

There was moderate heterogeneity between the studies ($I^2$ 42.9%). Baujat's plot indicated that one Korean study unduly influenced heterogeneity as well as pooled estimates [42]. No additional influential study was identified on formal influence analysis. Omitting this single study from analysis resulted in a lower summary RR estimate (1.35, 95% CI 0.98–1.87) and lesser heterogeneity ($I^2$ 29.5%). On sensitivity analysis, omitting other studies one at a time did not significantly affect heterogeneity (S4 Fig). On subgroup analysis, studies conducted in Africa or in low tuberculosis burden or multiple countries, as well as population-based studies, showed negligible heterogeneity (Table 3). Overall, the pooled RR estimates were much higher from studies conducted in countries not having high tuberculosis burden, as well as from population-based studies (Table 3). There was no significant publication bias (S5 Fig).

**Table 2. Subgroup analysis for summary estimates for proportion of COVID-19 patients with active pulmonary tuberculosis.**

| Criteria and subgroups | | No. of studies | Summary proportion, % (95% CI) | $I^2$, % |
|---|---|---|---|---|
| Overall | | 43 | 0.99 (0.74–1.27) | 94.7 |
| Continent: | Africa | 10 | 1.31 (0.73–2.04) | 98.4 |
| | Asia | 30 | 1.13 (0.81–1.49) | 83.5 |
| | Other/Multiple countries | 3 | 0.37 (0.02–1.02) | - |
| Study design: | Prospective | 4 | 1.49 (0.47–2.98) | 68.1 |
| | Not prospective | 39 | 1.04 (0.78–1.34) | 95.1 |
| Study setting: | Hospital-based | 37 | 1.34 (0.94–1.80) | 84.5 |
| | Population-based | 6 | 0.71 (0.35–1.19) | 99.1 |
| Patient inclusion: | Confirmed cases only | 40 | 1.03 (0.77–1.33) | 94.9 |
| | Probable cases also | 3 | 1.80 (0.41–4.01) | - |
| Tuberculosis definition: | Criteria specified | 15 | 0.95 (0.60–1.38) | 97.9 |
| | Criteria not specified | 28 | 1.21 (0.82–1.65) | 75.1 |
| Tuberculosis burden: | High burden countries | 39 | 1.24 (0.94–1.57) | 94.5 |
| | Other/multiple countries | 4 | 0.30 (0.15–0.49) | 72.6 |
| Study quality: | NOS score > = 7 | 6 | 0.83 (0.43–1.35) | 95.7 |
| | NOS score <7 | 37 | 1.19 (0.87–1.56) | 94.2 |

95% CI 95% confidence interval, $I^2$ Higgins' inconsistency index, NOS Newcastle-Ottawa Scale for study quality.

## Need for hospitalization

Four publications with 28,438 COVID-19 patients, of whom 479 (1.7%) had tuberculosis, provided data on hospitalization due to COVID-19 [36, 45, 52, 71]. All these studies were from high tuberculosis burden countries (two from South Africa, and one each from China and Philippines), had a retrospective study design, and included patients with laboratory confirmed COVID-19. Two (50.0%) studies were of high quality [36, 52]. Overall, 20.6% of patients were hospitalized. Of the 5853 patients who required hospitalization in the included cohorts, 227 (3.9%) had underlying tuberculosis. Two studies reported a RR for hospitalization that statistically significantly exceeded 1.0 (Fig 3) [36, 71]. COVID-19 patients who also had tuberculosis were 1.86 (95% CI 0.91–3.81) times more likely require hospitalization as compared to COVID-19 patients without tuberculosis (Fig 3). This pointed to the absence of any statistically significant risk of hospitalization among COVID-19 patients with tuberculosis.

There was considerable heterogeneity between the studies ($I^2$ 97.5%). A subgroup analysis was not undertaken due to small number of studies. There was no significant publication bias (S5 Fig).

## Mortality

Seventeen studies with 42,321 COVID-19 patients, of whom 632 (1.5%) had tuberculosis, reported on deaths due to COVID-19 [36, 39, 41–43, 45, 47–50, 52, 57, 60, 63, 65–67]. All studies were conducted in high tuberculosis burden countries, except one from South Korea and another that combined data from multiple nations [42, 43]. All studies, except one, included patients with laboratory confirmed COVID-19 [39]. Only one publication had a prospective study design [39]. Four (23.5%) studies were considered as high quality [36, 42, 43, 52]. Of the 2822 patients who died in the included cohorts, 97 (3.4%) had underlying tuberculosis. Only four (23.5%) studies reported RR for mortality that clearly exceeded 1.0 (Fig 3) [36, 42, 45, 52]. The confidence limits for all other studies were wide (Fig 3). COVID-19 patients who also had

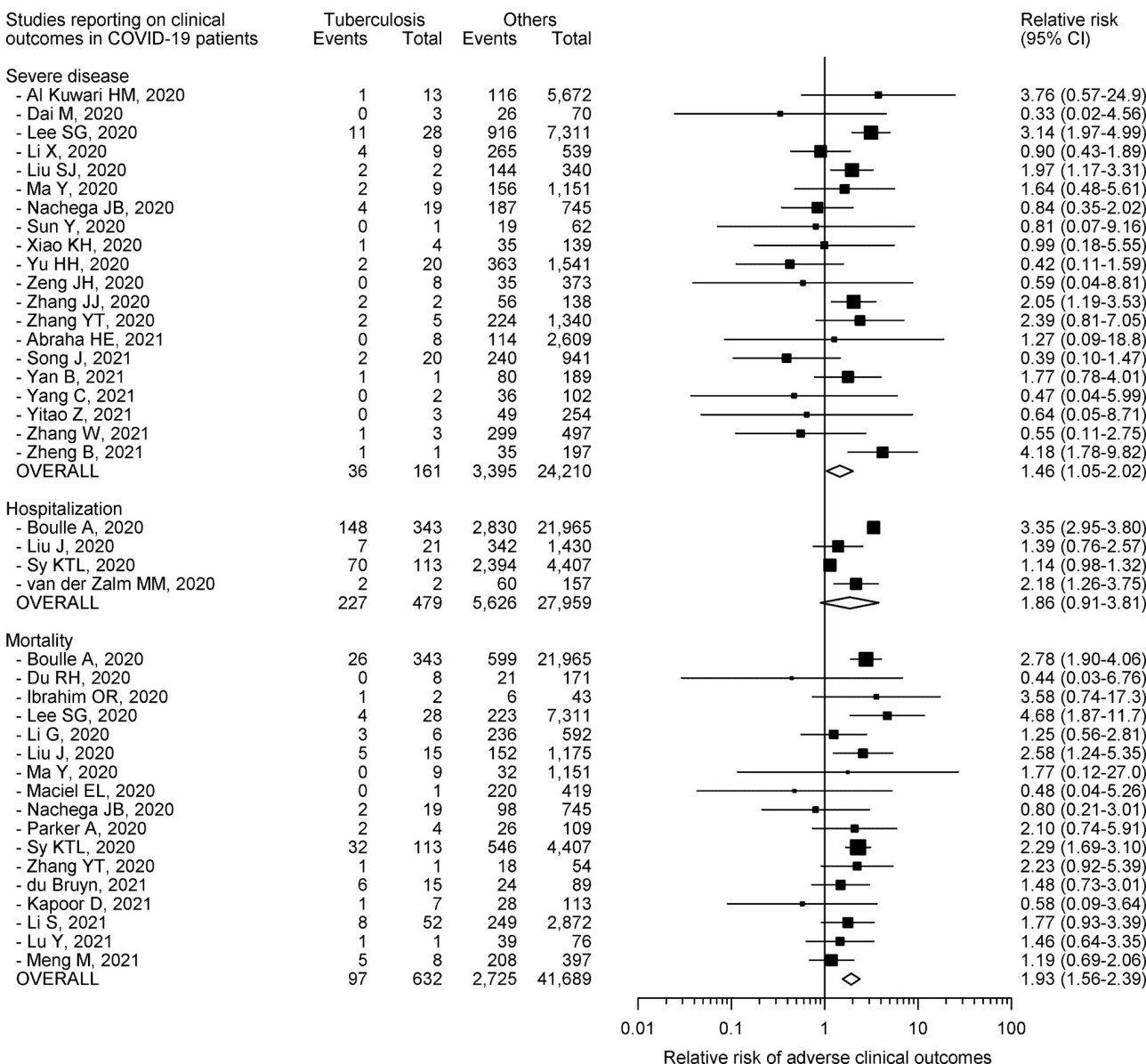

**Fig 3. Relative risk, and corresponding 95% confidence intervals (CI), of adverse clinical outcomes among COVID-19 patients having tuberculosis.**

tuberculosis were 1.93 (95% CI 1.56–2.39) times more likely to die as compared to COVID-19 patients without tuberculosis (Fig 3).

There was only mild heterogeneity between the studies ($I^2$ 21.5%). Inspection of Baujat's plot indicated that four studies unduly affected heterogeneity as well as pooled estimates [36, 42, 52, 67]. Two of these were also considered to be influential on formal influence analysis [52, 67]. Omitting these four studies from analysis lowered the summary RR estimate (1.65%, 95% CI 1.25–2.18) and resulted in negligible heterogeneity ($I^2$ 0.0%). On sensitivity analysis, heterogeneity could also be further decreased by individually omitting three of these studies one at a time (S4 Fig) [36, 42, 67]. Stratification by whether criteria for tuberculosis definition were specified in the studies resulted in homogeneity in either group (Table 3). On subgroup analysis, studies conducted outside Asia or Africa or in low tuberculosis burden or multiple

**Table 3. Subgroup analysis for summary relative risk of adverse outcomes among COVID-19 patients with active pulmonary tuberculosis.**

| Criteria and subgroups | | No. of studies | COVID-19 severity Summary relative risk (95% CI) | $I^2$, % | No. of studies | Mortality Summary relative risk (95% CI) | $I^2$, % |
|---|---|---|---|---|---|---|---|
| Overall | | 20 | 1.46 (1.05–2.02) | 42.9 | 17 | 1.93 (1.56–2.39) | 21.5 |
| Continent: | Africa | 2 | 0.87 (0.38–2.01) | 0.0 | 5 | 2.10 (1.38–3.18) | 24.7 |
| | Asia | 18 | 1.52 (1.08–2.15) | 42.9 | 10 | 1.94 (1.47–2.56) | 23.1 |
| | Other/Multiple countries | - | - | - | 2 | 1.14 (0.53–2.44) | 0.0 |
| Study design: | Prospective | 1 | 2.05 (1.19–3.53) | - | 1 | 0.44 (0.03–6.76) | - |
| | Not prospective | 19 | 1.37 (0.95–1.98) | 45.3 | 16 | 1.93 (1.57–2.41) | 21.5 |
| Study setting: | Hospital-based | 18 | 1.32 (0.95–1.82) | 30.9 | 14 | 1.66 (1.25–2.22) | 14.4 |
| | Population-based | 2 | 3.17 (2.02–4.97) | 0.0 | 3 | 2.37 (1.90–2.96) | 0.0 |
| Patient inclusion: | Confirmed cases only | 18 | 1.44 (0.98–2.11) | 43.1 | 16 | 1.93 (1.56–2.39) | 21.5 |
| | Probable cases also | 2 | 1.39 (0.65–2.97) | 65.0 | 1 | 0.44 (0.03–6.76) | 0.0 |
| Tuberculosis definition: | Criteria specified | 5 | 1.46 (0.64–3.33) | 72.3 | 6 | 2.58 (2.08–3.20) | 0.0 |
| | Criteria not specified | 15 | 1.50 (1.06–2.11) | 23.7 | 11 | 1.41 (1.08–1.84) | 0.0 |
| Tuberculosis burden: | High burden countries | 18 | 1.32 (0.95–1.82) | 30.9 | 15 | 1.94 (1.59–2.38) | 11.4 |
| | Other/multiple countries | 2 | 3.17 (2.02–4.97) | 0.0 | 2 | 2.38 (0.65–8.66) | 77.7 |
| Study quality: | NOS score > = 7 | 3 | 1.31 (0.29–5.97) | 75.3 | 4 | 2.44 (1.75–3.41) | 42.3 |
| | NOS score <7 | 17 | 1.45 (1.05–1.99) | 27.0 | 13 | 1.58 (1.22–2.04) | 0.0 |

95% CI 95% confidence interval, $I^2$ Higgins' inconsistency index, NOS Newcastle-Ottawa Scale for study quality.

countries, population-based studies, those including patients with a clinico-radiological diagnosis of COVID-19, and low-quality publications showed negligible heterogeneity (Table 3). There was no significant publication bias (S5 Fig).

## Discussion

We found that 0.99% of the COVID-19 patients had active pulmonary tuberculosis. These patients showed higher risk for mortality, but not for severe disease or hospitalization, than COVID-19 patients without tuberculosis. Our data synthesis summarizes far greater number of studies than previous meta-analyses and provides information both on tuberculosis frequency and COVID-19 outcome estimates [12, 19, 20]. Unlike previous meta-analyses that reported summary odds ratios, we present summary RR estimates for adverse clinical outcomes, which are much easier to interpret and understand in a clinical setting.

The summary proportion of those with active pulmonary tuberculosis among COVID-19 patients appears higher than the recent WHO estimates for annual incidence of tuberculosis in some of the high tuberculosis burden countries where most of the studies were conducted (China 0.06%, India 0.19%, Nigeria 0.22%, Philippines 0.55%, and South Africa 0.61%) [31]. However, this proportion of active tuberculosis is lower than the generally reported proportion of other comorbid conditions, like diabetes or hypertension [5, 8]. Whether the lower

tuberculosis proportion is due to under-reporting or under-recognition of active tuberculosis among COVID-19 patients, or to safeguarding strategies commonly employed by people with respiratory disorders, is not certain. However, when patients with active pulmonary tuberculosis do acquire COVID-19, there is a significantly greater risk (about two-fold higher) of COVID-19 mortality. Our summary estimate for relative risk of mortality in COVID-19 patients having tuberculosis is quite similar to the relative risk estimates of mortality for COVID-19 patients having other comorbid conditions (like diabetes, hypertension, or cardiovascular diseases) widely known to adversely affect prognosis in COVID-19 patients [1–3]. It is likely that superadded COVID-19 pneumonia in a lung that is already structurally damaged by tuberculosis may manifest as more severe disease. Importantly, local alterations in lung immunity resulting from active pulmonary tuberculosis can also adversely influence host response to SARS-CoV-2 virus. Recent in-vitro data from COVID-19 patients with active pulmonary tuberculosis has shown an attenuated interferon-gamma response after stimulation of whole blood with peptides derived from SARS-CoV-2 spike protein, in contrast to a normal response to *Mycobacterium tuberculosis*-specific antigens [78].

There are several similarities between COVID-19 and pulmonary tuberculosis. In several countries, COVID-19 too is a stigmatizing disorder, much like tuberculosis. Both diseases show airborne transmission when people are in close contact. Both present with similar symptoms like fever and cough. This can complicate decision-making, especially is nations with high tuberculosis burden. Although several countries have proposed bidirectional screening of both COVID-19 and pulmonary tuberculosis patients, such policy remains difficult to implement in resource-constrained settings. This might contribute to underdiagnosis of tuberculosis in COVID-19 patients. As it is, under-reporting of tuberculosis is a problem that is globally recognized. This is further compounded by reduced access to tuberculosis diagnosis and treatment as a result of COVID-19 related restrictions. It is therefore possible that our calculations regarding pulmonary tuberculosis among COVID-19 patients might be an underestimate.

Our systematic review has a few limitations. Due to the dynamic nature of the pandemic, and the lag between data collection and publication of results, most studies provide information from the initial months of 2020 and from regions that were severely afflicted earlier. Thus, the figures may not truly represent the patient data from all the geographic locations. Also, most of the included studies had a retrospective design, and collated data from medical records that were likely completed in an overwhelmed health system. This could have resulted in both underreporting as well as misclassification of comorbid health conditions. Several studies reported only on inpatients who have a higher probability of adverse outcomes compared to patients in the community. Only 15.6% of the included studies were of sufficiently high quality. There were differences in healthcare strategies regarding SARS-CoV-2 testing and admission/transfer criteria, variability in institutional practices in the timing of investigations and other evaluations, and the level and extent of medical intervention available to patients. Such heterogeneity can restrict the generalizability of our results. We cannot rule out an overestimation from lack of adjustment for potential confounders (like age, HIV status, other comorbid health conditions, or other patient characteristics) as we focused on univariate estimates. In particular, only one South African study reported on tuberculosis frequency data and outcome parameters stratified by HIV status, and there is need to gather more information on the impact of HIV on COVID-19 and tuberculosis associations.

## Conclusion

In summary, the available evidence suggests that COVID-19 patients show relatively higher proportion of concurrent active pulmonary tuberculosis. Active pulmonary tuberculosis significantly increases the risk of severe COVID-19 and COVID-19-related mortality.

## Supporting information

**S1 Table. Details of Newcastle-Ottawa Scale scoring for study quality.**
(PDF)

**S1 Fig. Baujat's plot for studies reporting prevalence of tuberculosis among COVID-19 patients.**
(PDF)

**S2 Fig. Influence statistics for studies reporting prevalence of tuberculosis among COVID-19 patients.**
(PDF)

**S3 Fig. Sensitivity analysis for studies reporting prevalence of tuberculosis among COVID-19 patients.**
(PDF)

**S4 Fig. Sensitivity analysis for studies reporting outcomes of patients COVID-19 patients with comorbid tuberculosis.**
(PDF)

**S5 Fig. Contour-enhanced trim-and-fill funnel plots.**
(PDF)

## Author Contributions

**Conceptualization:** Ashutosh Nath Aggarwal.

**Data curation:** Ashutosh Nath Aggarwal, Ritesh Agarwal.

**Formal analysis:** Ashutosh Nath Aggarwal, Ritesh Agarwal, Sahajal Dhooria, Kuruswamy Thurai Prasad, Inderpaul Singh Sehgal, Valliappan Muthu.

**Methodology:** Ashutosh Nath Aggarwal, Ritesh Agarwal, Sahajal Dhooria, Kuruswamy Thurai Prasad, Inderpaul Singh Sehgal, Valliappan Muthu.

**Project administration:** Ashutosh Nath Aggarwal.

**Supervision:** Ashutosh Nath Aggarwal, Ritesh Agarwal.

**Validation:** Ashutosh Nath Aggarwal, Ritesh Agarwal, Sahajal Dhooria, Kuruswamy Thurai Prasad, Inderpaul Singh Sehgal, Valliappan Muthu.

**Visualization:** Ashutosh Nath Aggarwal, Ritesh Agarwal, Sahajal Dhooria, Kuruswamy Thurai Prasad, Inderpaul Singh Sehgal, Valliappan Muthu.

**Writing – original draft:** Ashutosh Nath Aggarwal, Ritesh Agarwal, Sahajal Dhooria, Kuruswamy Thurai Prasad, Inderpaul Singh Sehgal, Valliappan Muthu.

**Writing – review & editing:** Ashutosh Nath Aggarwal, Ritesh Agarwal, Sahajal Dhooria, Kuruswamy Thurai Prasad, Inderpaul Singh Sehgal, Valliappan Muthu.

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
