## [Decision Letter · Decision Letter 0]

25 Aug 2021

PONE-D-21-23415

Active pulmonary tuberculosis and coronavirus disease 2019: a systematic review and meta-analysis

PLOS ONE

Dear Dr. Aggarwal,

Thank you for submitting your manuscript to PLOS ONE. After careful consideration, we feel that it has merit but does not fully meet PLOS ONE’s publication criteria as it currently stands. Therefore, we invite you to submit a revised version of the manuscript that addresses the points raised during the review process.

We look forward to receiving your revised manuscript.

Kind regards,

Girish Chandra Bhatt, MD, FASN

Academic Editor

PLOS ONE

Journal Requirements:

2. Please confirm that you have included all items recommended in the PRISMA checklist including:

- the full electronic search strategy used to identify studies with all search terms and limits for at least one database.

- a Supplemental file of the results of the individual components of the quality assessment, not just the overall score, for each study included.

- an assessment of publication bias using graphical methods (e.g. Funnel plot) and statistical methods (e.g. Egger’s test) as appropriate

- See https://journals.plos.org/plosmedicine/article?id=10.1371/journal.pmed.1000100#pmed-1000100-t003 for guidance on reporting.

Additional Editor Comments:

The systematic review is performed with robust methodology. The authors should revise the manuscript as per the suggestions by reviewers. I have few more suggestions to further improve the manuscript:

1) As there is unexplained heterogeneity (for proportion of the patients with active TB) the authors should use Bajaut's curve to identify the outlier studies and perform an influential analysis. Also, leave one out is also recommended in case of high heterogeneity.

2) Wherever, the pooled results are given, authors should also give I2 stats.

Reviewers' comments:

Reviewer's Responses to Questions

**Comments to the Author**

1. Is the manuscript technically sound, and do the data support the conclusions?

Reviewer #1: Partly

Reviewer #2: Yes

2. Has the statistical analysis been performed appropriately and rigorously? 

Reviewer #1: Yes

Reviewer #2: Yes

3. Have the authors made all data underlying the findings in their manuscript fully available?

Reviewer #1: No

Reviewer #2: Yes

4. Is the manuscript presented in an intelligible fashion and written in standard English?

Reviewer #1: Yes

Reviewer #2: Yes

5. Review Comments to the Author

Reviewer #1: 1. Please provide at least search strategy for Pubmed/Medline as free text search has a chance to miss out some of the studies that may have important bearing on the findings. Moreover, search strategy should be reproducible (it is like participants in a study, and any wrong selection method may lead to selection bias.

2. We explored the reasons for heterogeneity only if data from ten or more studies were summarized for any outcome: why 10 studies ?? any particular reason (if yes, may provide citation)

3. For searching the reasons for heterogeneity, we performed subgroup analyses: Is it sub-group or sensitivity analysis ?

4. The authors found that TB is increasing mortality but not severity of Covid 19..What is the explanation for this ?

5. The figures are of poor quality

Reviewer #2: This is the first real comprehensive and systematic analysis on how active tuberculosis (TB) infection impacts COVID-19 outcomes, including severity of disease, hospitalization, and mortality. This is a well written manuscript that covers an important area of research that has not been addressed in the literature in such a systematic way. That said, I still propose some minor ways to revise and strengthen the current manuscript.

Revisions

1. Line 76-80: The sentence begins by describing “TB patients with COVID-19” and ends with “COVID-19 patients without TB”. The language must be consistent and accurate, throughout the entire manuscript. In this case, you must be clear that you are referring to “COVID-19 patients with/without TB”, and not the other way around (the interpretation of the data would be completely different). Again, please check the language throughout the entire manuscript.

2. Line 96-97: COVID manifests in different ways for different people, many of whom are asymptomatic. I do not think it is accurate to say COVID-19 often manifests as severe pneumonia, especially without a supporting reference.

3. Line 181: Add a comma for long numbers to make it more legible.

4. Line: 202: Why were two studies that did not report patients with active TB included? In the PRISMA diagram, it shows that studies were excluded because there was no data on TB. All this to say, these 2 papers (Krati 2020, Gidado M, 2020) do not seem relevant to this meta-analysis and I do not think they should be included as they may bias the results.

5. Lie 208: Because you are talking about two diseases (TB and COVID), you must pay attention to which disease you are referring to. In this line, “low burden” is probably referring to TB, but this may not be clear to all readers. Please specific “low TB/COVID burden” when relevant throughout the paper.

6. Line 212-213: Might be helpful to include an appendix that have the Egger’s test graphs for the readers to evaluate the publication bias themselves.

7. Line 215: There needs to be some way of describing what is meant by ‘sever COVID-19’. Was this a definition that the studies used? Was it the same across studies? If not, how did you decide to combine studies with different definitions of COVID-19 severity? Not clear

8. Line 225-229: When talking about quantitative data in the text it’s usually good to include it in the text so the reader doesn’t have to go look for the data right away. This is regarding the sub-group heterogeneity and the pooled RRs mentioned in this section.

9. Line 250: 6.7% mortality rate for who? All COVID-19 patients? COVID-19 patients with TB? Not clear

10. Line 253-254: General comment about describing results. Similar to my first point, you must be very clear with the interpretation of the RR in this paper. For example, I would say: “people with COVID who also had TB were 1.93 (CI XX) times more likely to die compared to people with COVID who didn’t have TB”. Make sure this interpretation is applied to all RR results.

11. Line 263-269: Since only one study stratified by HIV status, it doesn’t seem relevant to repeat the results here. This just seems like you are re-iterating the results from another study instead of presenting a new analysis. Suggest removing this section entirely and discussing the importance of segregating by HIV status/acknowledge one on paper did this in the discussion.

12. Line 285-287: This is a very important point that I think needs further elaboration. I think there is evidence in the literature that TB cases have been severely underreported. I think there is also evidence that the overlapping symptoms between COVID and TB complicate clinical decision making. The lack of bi-directional screening further complicates this matter. I think this should be highlighted in the discussion further as it is very important.

13. Line 289: Which summary estimate are you referring to?

14. Line 291-293: are these alterations in lungs due to TB?

6. PLOS authors have the option to publish the peer review history of their article (what does this mean?). If published, this will include your full peer review and any attached files.

Reviewer #1: No

Reviewer #2: No

---

## [Author Response · Author response to Decision Letter 0]

22 Sep 2021

PONE-D-21-23415

Active pulmonary tuberculosis and coronavirus disease 2019: a systematic review and meta-analysis

POINTWISE REPLIES TO COMMENTS

Journal Requirements:

This has been ensured.

2. Please confirm that you have included all items recommended in the PRISMA checklist including:

- the full electronic search strategy used to identify studies with all search terms and limits for at least one database.

- a Supplemental file of the results of the individual components of the quality assessment, not just the overall score, for each study included.

- an assessment of publication bias using graphical methods (e.g. Funnel plot) and statistical methods (e.g. Egger’s test) as appropriate

The full search strategy has been provided for PubMed database in the Methods section (lines 127-130). S1 Table (online supplement) details the individual NOS score components for each included study. Additional details on formal statistical and graphical evaluation of bias are now included in text, and funnel plots have been provided as a supplemental file (S5 Fig).

Additional Editor Comments:

1) As there is unexplained heterogeneity (for proportion of the patients with active TB) the authors should use Baujat's curve to identify the outlier studies and perform an influential analysis. Also, leave one out is also recommended in case of high heterogeneity.

All these additional statistical procedures have been incorporated into the manuscript text (lines 170-171, 176-180, 214-223, 243-248, 281-286), and graphical details provided as supplemental files (S1 Fig, S2 Fig, S3 Fig, and S4 Fig).

2) Wherever, the pooled results are given, authors should also give I2 stats.

I2 values are provided throughout in the text wherever pooled estimates are given.

Reviewer #1: 

1. Please provide at least search strategy for Pubmed/Medline as free text search has a chance to miss out some of the studies that may have important bearing on the findings. Moreover, search strategy should be reproducible (it is like participants in a study, and any wrong selection method may lead to selection bias.

The actual search string used to query the PubMed database is now provided in the Methods section (lines 127-130).

2. We explored the reasons for heterogeneity only if data from ten or more studies were summarized for any outcome: why 10 studies ?? any particular reason (if yes, may provide citation)

There is no clear guidance on the minimum number of studies needed to explore heterogeneity. The Cochrane handbook suggests 10 studies in each group for performing subgroup analysis or meta-regression. Fu et al (J Clin Epidemiol 2011;64:1187-97) have proposed that each categorical subgroup should have a minimum of 4 studies. While Geissbuhler et al (BMC Med Res Methodol 2021;21:123) recently recommend at least five studies per group. We considered analysing multiple subgroups, each with a potentially variable number of studies, and arbitrarily decided beforehand on a minimum figure of ten studies (total) before undertaking subgroup analysis. We may also point out that we ultimately performed subgroup analysis for all summary estimates, except for hospitalization as outcome (four studies only), based on this arbitrary number. However, we understand that our choice was at best subjective, and have therefore removed this sentence from the manuscript. For the hospitalization outcome section in the Results, we have already stated that subgroup analysis was not performed due to extremely few studies.

3. For searching the reasons for heterogeneity, we performed subgroup analyses: Is it sub-group or sensitivity analysis?

We performed subgroup analyses that focused on differential effects between distinct subgroups, rather than on separate effects in different analyses. We now also present results on sensitivity analysis (leave-one-study-out analysis).

4. The authors found that TB is increasing mortality but not severity of Covid 19. What is the explanation for this ?

We apologize that due to a typographical error, we reported erroneous values for relative risk for severe COVID-19, both in the text and the Forest plot. The error has now been rectified (lines 240-242) and the correct figures suggest that COVID-19 patients with tuberculosis are at a significantly increased risk of severe COVID-19.

5. The figures are of poor quality.

We have submitted figures at 600 dpi resolution. We have also passed all three figures through the Preflight Analysis and Conversion Engine (PACE) digital diagnostic tool to ensure they meet PLOS requirements.

Reviewer #2: 

1. Line 76-80: The sentence begins by describing “TB patients with COVID-19” and ends with “COVID-19 patients without TB”. The language must be consistent and accurate, throughout the entire manuscript. In this case, you must be clear that you are referring to “COVID-19 patients with/without TB”, and not the other way around (the interpretation of the data would be completely different). Again, please check the language throughout the entire manuscript.

We apologise for the inconsistency and understand that this may result in an inaccurate understanding of our results. The same has been checked and corrected throughout the manuscript (lines 76-77, 240-242, 262-263, 278-280).

2. Line 96-97: COVID manifests in different ways for different people, many of whom are asymptomatic. I do not think it is accurate to say COVID-19 often manifests as severe pneumonia, especially without a supporting reference.

We agree and have removed the phrase from Introduction.

3. Line 181: Add a comma for long numbers to make it more legible.

As suggested, a comma has been added as a thousand’s separator throughout the manuscript text, as well as in tables and figures.

4. Line: 202: Why were two studies that did not report patients with active TB included? In the PRISMA diagram, it shows that studies were excluded because there was no data on TB. All this to say, these 2 papers (Krati 2020, Gidado M, 2020) do not seem relevant to this meta-analysis and I do not think they should be included as they may bias the results.

As suggested, we have removed these two studies from our meta-analysis and have re-calculated our summary estimates accordingly.

5. Line 208: Because you are talking about two diseases (TB and COVID), you must pay attention to which disease you are referring to. In this line, “low burden” is probably referring to TB, but this may not be clear to all readers. Please specific “low TB/COVID burden” when relevant throughout the paper.

As suggested, the necessary corrections have been made wherever “low burden” meant “low TB burden” (lines 224, 289, 311).

6. Line 212-213: Might be helpful to include an appendix that have the Egger’s test graphs for the readers to evaluate the publication bias themselves.

We have included funnel plots, and presented data on Egger’s test, in a supplemental file (S5 Fig).

7. Line 215: There needs to be some way of describing what is meant by ‘severe COVID-19’. Was this a definition that the studies used? Was it the same across studies? If not, how did you decide to combine studies with different definitions of COVID-19 severity? Not clear

There is no standard definition for severe COVID-19. Stratification into severe or non-severe disease is based on a combination of several parameters that include respiratory rate, oxygenation, and mode of respiratory support, etc. We therefore defined severe COVID-19 based on use of institutional or national guidelines, or guidance from international professional bodies or the World Health Organization, as chosen by individual authors. Most such criteria have only minor differences in the way COVID-19 severity is categorized, and hence we pooled all such data under a single umbrella grouping of severe COVID-19. This is now mentioned in the Methods section (lines 146-147), and further elaborated in the severe COVID-19 section of the Results (lines 232-236).

8. Line 225-229: When talking about quantitative data in the text it’s usually good to include it in the text so the reader doesn’t have to go look for the data right away. This is regarding the sub-group heterogeneity and the pooled RRs mentioned in this section.

We understand that it is simple for the reader to have the numbers in the text, rather than a separate table placed remotely in the publication, if only limited data is presented. In the current case, we would need to present more than half the table in the text, with some tables needing citation more than once. This would unnecessarily confuse the reader, while a table puts everything together at one place for easy reference and comparison. We therefore request the honorable reviewer to allow us to continue with the present scheme. We have already summarized the important observations from the tables (with references as appropriate).

9. Line 250: 6.7% mortality rate for who? All COVID-19 patients? COVID-19 patients with TB? Not clear.

This sentence has been deleted.

10. Line 253-254: General comment about describing results. Similar to my first point, you must be very clear with the interpretation of the RR in this paper. For example, I would say: “people with COVID who also had TB were 1.93 (CI XX) times more likely to die compared to people with COVID who didn’t have TB”. Make sure this interpretation is applied to all RR results.

We agree, and have modified all such statements as suggested (lines 76-77, 240-242, 262-263, 278-280).

11. Line 263-269: Since only one study stratified by HIV status, it doesn’t seem relevant to repeat the results here. This just seems like you are re-iterating the results from another study instead of presenting a new analysis. Suggest removing this section entirely and discussing the importance of segregating by HIV status/acknowledge one on paper did this in the discussion.

As suggested, we have removed the entire section on HIV status from our results and have simply mentioned this while describing characteristics of included studies in the first paragraph of Results section (lines 202-203). We have also added this as a limitation of our meta-analysis, and outlined the need for additional information, in the last paragraph of our Discussion (lines 353-355).

12. Line 285-287: This is a very important point that I think needs further elaboration. I think there is evidence in the literature that TB cases have been severely underreported. I think there is also evidence that the overlapping symptoms between COVID and TB complicate clinical decision making. The lack of bi-directional screening further complicates this matter. I think this should be highlighted in the discussion further as it is very important.

We have highlighted these points in Discussion as a new paragraph (lines 328-338).

13. Line 289: Which summary estimate are you referring to?

We meant summary estimate for relative risk of mortality in COVID-19 patients having tuberculosis. This has now been elaborated (lines 318-320).

14. Line 291-293: are these alterations in lungs due to TB?

Yes. This has now been more explicitly stated in second paragraph of Discussion (lines 322-324).

---

## [Decision Letter · Decision Letter 1]

11 Oct 2021

Active pulmonary tuberculosis and coronavirus disease 2019: a systematic review and meta-analysis

PONE-D-21-23415R1

Dear Dr. Aggarwal,

We’re pleased to inform you that your manuscript has been judged scientifically suitable for publication and will be formally accepted for publication once it meets all outstanding technical requirements.

Kind regards,

Girish Chandra Bhatt, MD, FASN

Academic Editor

PLOS ONE

Additional Editor Comments (optional):

Reviewers' comments:

Reviewer's Responses to Questions

**Comments to the Author**

1. If the authors have adequately addressed your comments raised in a previous round of review and you feel that this manuscript is now acceptable for publication, you may indicate that here to bypass the “Comments to the Author” section, enter your conflict of interest statement in the “Confidential to Editor” section, and submit your "Accept" recommendation.

Reviewer #1: All comments have been addressed

2. Is the manuscript technically sound, and do the data support the conclusions?

Reviewer #1: Yes

3. Has the statistical analysis been performed appropriately and rigorously? 

Reviewer #1: Yes

4. Have the authors made all data underlying the findings in their manuscript fully available?

Reviewer #1: Yes

5. Is the manuscript presented in an intelligible fashion and written in standard English?

Reviewer #1: Yes

6. Review Comments to the Author

Reviewer #1: Dear authors

Thank you for your response. I could find that all my concerns have been addressed satisfactorily. I dont have any further comments to make.

Thank you

7. PLOS authors have the option to publish the peer review history of their article (what does this mean?). If published, this will include your full peer review and any attached files.

Reviewer #1: **Yes: **Rashmi Ranjan Das

---

## [Editor Report · Acceptance letter]

13 Oct 2021

PONE-D-21-23415R1 

Active pulmonary tuberculosis and coronavirus disease 2019: a systematic review and meta-analysis 

Dear Dr. Aggarwal:

I'm pleased to inform you that your manuscript has been deemed suitable for publication in PLOS ONE. Congratulations! Your manuscript is now with our production department. 

Kind regards, 

on behalf of

Dr. Girish Chandra Bhatt 

Academic Editor

PLOS ONE